# Factors Associated with Communities’ Satisfaction with Receiving Curative Care Administered by Community Health Workers in the Health Districts of Bousse and Boussouma in Burkina Faso, 2024

**DOI:** 10.3390/ijerph21091151

**Published:** 2024-08-30

**Authors:** Hamed Sidwaya Ouedraogo, Abdoul-Guaniyi Sawadogo, Ahmed Kabore, Badra Ali Traore, Mamadou Traore, Massoudou Harouna Maiga, Marcella Valerie Zombre Sanon, Maxime Koine Drabo

**Affiliations:** 1Ministry of Health and Public Hygiene, Ouagadougou 03 BP 7009, Burkina Faso; madss07@outlook.fr (M.T.); anassimaiga@yahoo.fr (M.H.M.); marcellasanon@yahoo.fr (M.V.Z.S.); 2Public Health Laboratory of the UFR/SDS, Joseph KI-ZERBO University, Ouagadougou 03 BP 7021, Burkina Faso; ahmedkaboreza@gmail.com (A.K.); m_drabok@yahoo.fr (M.K.D.); 3Jhpiego Corporation, Ouagadougou 01 BP 5654, Burkina Faso; lesage2004bk@yahoo.fr; 4Institute of Sport Sciences and Human Development (ISSDH), Joseph KI-ZERBO University, Ouagadougou 03 BP 7021, Burkina Faso; 5Progress Consulting, Ouagadougou 09 BP 631, Burkina Faso; tba7@rocketmail.com; 6Health Sciences Research Institute (IRSS)/CNRST, Ouagadougou 03 BP 7192, Burkina Faso

**Keywords:** community health workers, community health care, quality, perception, Burkina Faso

## Abstract

Background: Since 2010, Burkina Faso has developed and initiated community-based management of childhood illnesses. Following the increased presence of community health workers and the adoption of free community health care, this study aims to assess community satisfaction with curative care administered by community health workers. Methodology: This was a descriptive and analytical cross-sectional study. Data were collected in the health districts of Boussé and Boussouma from 20 February to 30 March 2023 for quantitative data and from 12 to 30 January 2024 for qualitative data using a questionnaire (household survey) and an interview grid (focus groups). Analyses were conducted using SPSS IBM 25 and Nvivo 14. Results: Households benefit from oral curative care when using Community health workers, but are not satisfied with the temporal accessibility of these community health workers. Temporal accessibility and awareness during care have a significant influence on household satisfaction. Conclusions: Curative care by community health workers is effective, but its use could be improved by addressing the unavailability of community health workers, inputs and better communication during care.

## 1. Introduction

The global agenda of the 1980s was marked by the quest to reinforcing access to healthcare services close to where communities lived [1]. The concept of primary healthcare (PHC) was conceived, with the aim of improving community participation in managing their own health [2]. Since then, the concept has had mixed fortunes, with results that have sometimes been severely criticised [3], ranging from differing understandings among experts to incomplete implementation influenced by other agendas driven by some international institutions [4]. However, on balance, investment in strong primary care has been widely described as one of the most cost-effective and equitable ways [5] of moving towards universal health coverage [3,6]. Several countries, including Burkina Faso, then adopted and began implementing PHC and tried to operationalise it through several initiatives. With this in mind, most developing countries, especially those in sub-Saharan Africa, have used community health workers to develop community-based services, with a wide variety of programmes in place [7]. Community health workers (CHWs) are considered as the interface between the community (from which they come) and local health teams [8]. One of the key strategies implemented and based on community health workers is community-based integrated management of childhood illness (iCCM). It mainly targets killer childhood diseases such as malaria, pneumonia and diarrhoea. Several evaluations and studies have demonstrated the potential of this community-based management of childhood illness to reduce inequalities in access to care [9,10]. Burkina Faso has been piloting similar community healthcare programmes since the August 1983 revolution [11]. The country then implemented the iCCM strategy advocated by the World Health Organization (WHO) [12,13,14] with more extensive interventions using community health workers recruited within an institutionalised framework [15].

The iCCM was designed to improve geographical, financial and cultural access to healthcare for rural populations [14,16]. More specifically, its aim was to improve (i) early referral to the health centre when a child is ill, (ii) correct continuation of care at home by the health worker, and (iii) access to services through well-trained community relays in villages with no health centre [17]. It is now a priority tool for Burkina Faso in the fight against morbidity and mortality in children [18,19] and the development of a resilient healthcare system [20].

However, from 1989, when community-based management of children’s illnesses was very new, to recent times, the use of CHW services has remained inadequate according to several studies, resulting in low population coverage by iCCM in the areas where it has been implemented [10,21]. The various studies conducted have identified, among other things, the insufficient motivation of CHWs, frequent shortages of health products and insufficient coverage limiting the development of health service provision [12,13,17,22]. All these difficulties encountered in the implementation of the programme have probably affected people’s satisfaction with the curative care offered by the CHWs and have led to a low take-up of these services by the communities.

The majority of these studies were conducted before the reforms that led to the adoption of a harmonised profile of the CHWs in Burkina Faso, named the “Community-Based Health Worker (CBHWs)”, recruited and motivated by the Government of Burkina Faso with a well-organised package of activities. In 2016, the government adopted a free healthcare measure for children under five and extended community-based management of childhood illnesses in 2018 [23]. A new national community health strategy that included iCCM and a financial resource mobilisation package for said plan were prepared in 2018 and implemented [24]. We conducted this study in order to suggest actions that would improve the use of this care at a community level. It aims to investigate factors associated with community satisfaction in receiving curative care administered by health workers, so as to guide decision-making.

## 2. Materials and Methods

### 2.1. Type and Period of Study

This was a descriptive and analytical cross-sectional study, with data collected in the following two phases: from 20 February to 30 March 2023 (quantitative data) and from 12 to 30 January 2024 (qualitative data).

### 2.2. Study Framework

Burkina Faso is a developing country, according to the United Nations Development Programme (UNDP) [25]. The healthcare system is pyramidal, with three levels of care, the first of which is community-based and is intended to be the first point of contact for communities [26], with an average distance of 6.1 km to travel to access health centres [27].

The study was conducted in the health districts of Boussouma and Boussé in Burkina Faso. The Boussé health district (Central Plateau region) covers five municipalities, with a population of 199,999 and has 33 health facilities. The Boussouma health district (Centre Nord region) covers three rural municipalities, with a population of 239,894 and has 29 health facilities.

### 2.3. Study Population

The study focused on members of households (heads of households or guardians of children found during the study) and users of health facilities in Boussé and Boussouma health districts who agreed to participate.

### 2.4. Concept Approach, Techniques and Data Collection Tools

This study was conducted using the Beneguissé model for analysing the quality of care from the consumer’s perspective [28], comprising the following five dimensions: geographical accessibility, organisational accessibility, interpersonal communication, technical competence and continuity of care.

Quantitative data were collected using a questionnaire set up on smartphones using Kobotoolbox software. Prior to data collection, interviewers were trained for one day and the tools were pre-tested in health centres in the Bogodogo and Boulmiougou health districts. The interviewers were equipped with a dictaphone, and the speeches were transcribed. Qualitative data were collected using a semi-directive interview grid. The interviews were conducted face-to-face.

Data were collected on the basis of variables such as socio-demographic characteristics, the availability of human resources, the level of knowledge of household members, the level of satisfaction of beneficiaries with the services offered by CBHWs, the availability of resources, the existence of reference documents, geographical coverage and the quality of care offered by CBHWs. To operationalize the variable quality of care offered to children, we used a five-level Likert scale, representing “very satisfied”, “satisfied”, “fairly satisfied”, “not very satisfied” and “not at all satisfied”, and the quality of the tools was assessed using the Cronbach’s Alpha coefficient, which was 0.9.

### 2.5. Sampling

The study adopted various sampling techniques at different stages. Initially, two-stage cluster sampling was used. Given the exhaustive list of villages for the two districts, the villages to be surveyed were drawn without replacement. Households were then counted in each sampled village and the information entered. The list of households to be surveyed was generated automatically and the heads of household were then surveyed. Any household without a child under five was replaced.

For the qualitative part of the study, a simple random selection was made of one health facility per municipality; then, a village located more than 5 km away was chosen, where the security context allowed for data collection (some health facilities in these two districts are at risk of attack by armed groups). Focus groups of men and women of between five and ten people were conducted in these villages.

### 2.6. Data Treatment, Analysis and Ethical Aspects

The quantitative data were exported to Excel and cleaned and analysed using IBM SPSS version 25 software. In order to identify the aspects that have the greatest influence on household satisfaction with the services offered by the CBHWs, we performed a binary logistic regression to look for predictive factors [29].

The interviews were transcribed using Microsoft Word processing software, and those in national languages were translated before being transcribed. The qualitative data underwent a double analysis (inductive and deductive), and the elements of discourse linked to the themes relating to the variables of the study were retained.

The questionnaire included written consent from participants. The data were stored in a coded database with limited access.

This study was authorised by the National Health Research Ethics Committee of the Burkina Faso Ministry of Health (deliberation N°2023-03-061).

## 3. Results

### 3.1. Socio-Demographic Characteristics of Respondents

During this study, 960 heads of households living at least 5 km from a health facility agreed to answer our questionnaires. The average age of respondents was 35.37 years, with a standard deviation of 11.55. The average household size was 8.09, with a standard deviation of 4.26. Among our respondents, 64.8% (622/960) had not received any form of education. The majority were farmers by profession (50.7%, 487/960) and housewives (35%, 336/960).

Perceived quality of care was assessed in our study using the dimensions of satisfaction described by Beneguissé (Appendix A).

### 3.2. The Quality of CBHW Healthcare Services as Perceived by Households

#### 3.2.1. Geographical Accessibility

The average distance travelled by our respondents to reach the nearest health facility was 7.29 km (standard deviation 4.61). These households were well informed about the presence of CBHWs, according to the statements in Table 1.

The participants in our focus groups in these two districts describe CBHWs as villagers who are chosen, trained and then return to the village to care for the children. They said that they lived with them and expressed real satisfaction with the initiative. One participant told us: “*Such an initiative relieves us because there are times when a child is ill and we don’t even know how to get to the health centre because we don’t have any transportation”* Woman_R1_FG5.

Some respondents to the household survey were either not very satisfied or not at all satisfied (27.3%, 262/960) with geographical accessibility to care. Focus group participants said they had difficulty finding the CBHWs when their children were ill, as they were not at home (Table 1).

#### 3.2.2. The Dimension of Organisational Accessibility

The majority of households (99.5%, 955/960) claim to benefit from care administered by CBHWs when their children suffer from malaria, diarrhoea or pneumonia. In 99.8% of cases (958/960), they said they did not pay any fees to these community care providers. Table 2 shows the respondents’ level of satisfaction with the reception, overall waiting time and courtesy.

According to the focus group participants, these CBHWs visit them in winter to administer anti-malaria drugs as a preventive measure. They also reported receiving curative care. This satisfactory organisation of care is illustrated by what this respondent said: “*They visit us to check on our children’s health. During the rainy season, they go round our households to distribute medicines to our children, and when we see a sick child, they do a RDT to see if the child has malaria, and if the test is positive, they give us products to treat the child"* Woman_R3_FG18. The use of this curative care is limited by the lack of up-to-date information on the availability of stocks of medicines with CBHWs, and the difficulties in finding CBHWs, leading children’s parents to go first to the health facility (Table 1). Some villages have CBHWs that do not have any medicines and therefore refer households directly to the health centre, as one focus group participant put it: “*they say they don’t have all the medicines for treatment, so we have to go to the health centre”.* Woman_R6_FG11.

#### 3.2.3. The Dimension of Cultural Accessibility

According to the focus group participants, CBHWs are members of their communities (Table 1). They carry out a number of activities involving household awareness-raising on several topics, as summarised in Table 1, and this is much appreciated. Also, according to the results of the household survey, cultural accessibility (language, habits, etc.) and interaction between CBHWs and households during the administration of care are generally satisfactory (Table 2).

#### 3.2.4. The Technical Skills Dimension

CBHWs are much more likely to provide oral care with the administration of medicines and vitamins, according to the households surveyed (Table 3). However, measurement of the brachial perimeter, which is a prerequisite for nutritional management, was rarely mentioned by households, which did not quote it as a frequent type of care (Table 3). Of these households, 99.2% (952/960) said they trusted the CBHWs’ services. The effectiveness of curative care in some villages by CBHWs was confirmed by focus group participants, and illustrated by one of them as follows:

“With them we can get medicines and that saves us time because we don’t have to go to the health centre” Woman_R3_FG5.

However, several focus group participants raised the issue of the unavailability of products with CBHWs (Table 1). These gaps in the supply of medicines are illustrated by the following responses to the question “What is the first option when a child is ill?”:


*“In these situations, you have to bypass the CBHWs and go to the health centre, as they have nothing to treat you”*
Man_R4_FG14;


*“When a child is ill, I prefer to go directly to the health centre because the CBHWs doesn’t usually have all the products needed to treat the patient properly”*
Woman_R5_FG5

Some comments also show that households do not seek care early enough and no longer have a good perception of the technical capacity of CBHWs to provide care for their children (Table 1).

#### 3.2.5. Continuity of Care

Focus group participants described how CBHWs apply the referral system. Some households visit CBHWs, who refer them to the health facility according to the child’s clinical status. However, a large proportion of participants said they do not wait for the CBHWs’ advice and go directly to the health facilities, with the main reason for this being the unavailability of health products with CBHWs (Table 1).

### 3.3. Aspects Influencing Household Satisfaction and Confidence in CBHWs’ Care

From the associated bivariate analysis (Appendix A), household satisfaction with the care offered by the CBHWs was significantly associated with the level of education (X2 = 29, *p*-value = 0.000) and the temporal accessibility of care (X2 = 13.2, *p*-value = 0.010), which significantly influenced household satisfaction regarding the care offered by CBHWs at the 95% threshold.

Confidence in the care offered was significantly associated with vulnerability status (X2 = 35.3, *p*-value = 0.000), the reception by CBHWs (X2 = 15.1, *p*-value = 0.000) and cultural accessibility (X2 = 38.67, *p*-value = 0.002) at the 95% threshold.

### 3.4. Factors Predicting Household Satisfaction with CBHWs’ Care

Still using the components of satisfaction according to the conceptual model of the study, we progressively built a top-down “step-by-step” regression model by integrating all the explanatory variables into the model, then progressively removing the variables, which made it possible to build the final model.

All the conditions for use were checked beforehand (Appendix A), and the very high probability of X2 (*p*-value significant at 0.001 (<0.05)) shows that our regression model fits well.

Binary logistic regression enabled us to obtain results indicating that the elements of temporal accessibility to care (waiting time and planning of sessions) and the quality of awareness with practical advice during administration of care to the child (treatment of malaria, diarrhoea and pneumonia) contributed significantly to the prediction of household satisfaction in the use of care offered by CBHWs (Table 4).

## 4. Discussion

### 4.1. Use of CBHWs’ Care and Limiting Factors

The use of curative care offered by CBHWs varies from one health area to another. It is influenced by the availability of medicines and households’ level of information about the availability of medicine stocks. These reasons have already been mentioned by Thomas Druetz et al. [10], whose study on CHW usage in the health districts of Kaya and Zorgho identified the lack of information, the fact that people preferred the care offered at the health centre level, and drug stock-outs as the main reasons why people do not use these CHWs. The absence or inadequacy of CHBWs’ drug stocks is one of the difficulties that Seck et al. [12] and Ridde V. et al. [22] have also identified in the community management of childhood illnesses. This inadequacy may undermine the achievement of the results targeted by the implementation of iCCM. These drug stock-outs persist despite the extension of free care to community management of childhood illnesses decided in 2018 by the Government of Burkina Faso [23].

### 4.2. Perceived Quality According to Geographical and Temporal Accessibility to Care

We did not find any associations between household satisfaction and geographical accessibility. At first sight, this may seem contradictory to the conceptualisation of iCCM as a means of bringing care closer to households. The results obtained may be explained by the fact that these CHWs live in the villages with the households and that this is accepted by the households as a normal situation. Burkina Faso recruited 17,000 CBHWs [19,24,26,30,31,32] in 2016 through a bold policy by the Burkinabè government. This was later on supplemented by the recruitment of national voluntary CBHWs for urban areas and areas with security challenges [33]. However, the conclusions of Fletcher Njororai et al. [34] lead us to suggest adaptations for villages where concessions are scattered or where farming hamlets are located at significant distances from the village site where CBHWs are usually located. This could involve increasing the number of CBHWs or providing adequate transportation means to improve household satisfaction with CBHWs’ response times. This need to strengthen the geographical distribution of CBHWs and also to identify additional actions aimed at improving the quality of the care they offer to the most remote areas or areas facing security and/or humanitarian problems is necessary to strengthen equity, as reported by Champagne et al. [6] at the end of their study on community health worker (CHW) programmes.

### 4.3. Perception of the Time Accessibility and Technical Skills of CBHWs

Household satisfaction is influenced by the amount of time it said they preferred to go directly to the health facility, since CBHWs are often not immediately available when needed. And when they are often available, their role is limited to diagnosis, due to the lack or inadequacy of their stock of medicines. This is illustrated by respondents who say that they are referred to the health centre for treatment after their child’s illness has been diagnosed. According to Karen Leban et al. [8], the challenge of making medicines available determines the performance of CBHWs and undoubtedly has an impact on household satisfaction. CBHWs are full of technical skills that households appreciate, but the lack of medicines to treat them gives them a poor perception of their ability to provide adequate responses when they are faced with a sick child.

### 4.4. Interpersonal Communication

Households report that CBHWs are very active in communicating with them. This communication is very much appreciated by the respondents. This communication goes beyond iCCM and includes the preventive care offered to pregnant women, which was described as an excellent quality in the conclusions of the study by Danielle Burke et al. [13] because of the CBHWs’ mastery of the working tools available to them. These CBHWs sometimes have shortcomings, which were noted in the study by Fletcher Njororai et al. [34]. To remedy this, some authors such as Karen Leban et al. [8] have suggested strengthening their communication skills and providing up-to-date knowledge to deal with household enquiries. These suggestions are necessary in our context in order to improve confidence in CBHWs’ services and to improve perceived quality. This will be carried out in collaboration with health workers trained for this purpose over a few days, and will also build on opportunities for supervision, the whole of which will be implemented by including new technologies [35].

### 4.5. Continuity of Care

The aim of building this link in the system was to relieve the burden on health facilities and, above all, to reduce the distances households had to travel to health centres [14,17]. Burkina Faso has succeeded in setting up a healthcare system based on the stages for integrating community stakeholders that have already been identified in several previous studies. These include the recruitment, training and appointment of CBHWs, which Burkina Faso undertook in 2016 [13,15] with a remuneration system. This system was intended to be the first line of care with improved access, as prescribed by the iCCM implementation framework [14]. But the difficulties encountered have reduced the desired effects, and the notion of continuity of care is becoming difficult to achieve. Some of the households surveyed and focus group participants believed that recourse to CBHWs was useless. Efforts to resolve these difficulties are needed to maintain the overall perception of the usefulness of these CBHWs, whose role in early curative care could gradually fade away, despite the continuation of this policy of strengthening community care in Burkina Faso.

Strengthening the integration of services by reinforcing the implementation of nutrition activities during the community care of children and other related tasks could improve household perceptions of technical skills and the use of these services [36].

### 4.6. Factors Influenced Satisfaction and Confidence in CBHWs’ Care

Our study found that factors such as level of education and temporal accessibility significantly influenced household satisfaction with the care offered by CBHWs. Bartena Kimosop Samuel et al. [37] found that CHWs, in implementing community strategies, faced challenges such as being unemployed, having a business or earning a living through casual work. These professional situations had an impact on their performance. The difficulties of temporal accessibility found could be linked to the time given by CBHWs to their own professional activity, reducing their availability for their services. However, the time devoted to their work is associated with better performance in their services [38], which must necessarily influence the quality perceived by households. The professional activities performed by these CBHWs are imposed on them because of the low level of remuneration [15]. iCCM is extremely important and should be strengthened, given the association between vulnerability and trust in the care provided by CBHWs. Sonia Hamed et al. [16] insisted on the need to increase the resources allocated to the implementation of community interventions, in order to improve healthcare coverage for underserved populations. The perception of the quality of care provided by CBHWs in the security and humanitarian context of Burkina Faso will be improved and should strongly influence the use of CBHWs, who are responsible for the curative component of community health, one of the pillars of Burkina Faso’s health system resilience [20].

### 4.7. Factors Influencing Household Satisfaction

Household satisfaction is significantly influenced by time accessibility and awareness raising during the administration of medicines. These results indicate the need to work on the availability of CBHWs so that they can be accessed within a short timeframe and be the first option for households in the event of their children’s illness. The time taken to access this care was also mentioned in the study by Ayodele S. Jegede et al., in which CHWs were perceived as accessible and diligent [39]. The communication skills of stakeholders need to be strengthened by improving the guidelines for implementing communication activities by these CBHWs.

Also, in this communication component, there is a need to diversify the tools, to go beyond picture boxes and adopt audio and visual media in the languages spoken by these households. This will facilitate the acquisition of knowledge about the target iCCM diseases. This, in turn, can lead to better uptake of the care offered by the CBHWs. Also, in this digital era, thought should be given to finding a mechanism for using the communication platforms (e.g., digital media with animated images) that are best suited to the types of communication conducted on a daily basis by households. This will have an impact on the quality of CBHWs’ services and ultimately on their use.

Future studies could focus on the quality of communication during care and evaluate the communication tools available to CBHWs from the perspective of users. Action research could also test innovative tools that make maximum use of digital technology.

## 5. Conclusions

Burkina Faso has been implementing community health activities of varying scale for several decades. This mixed methods study was used to assess the quality of care provided by CBHWs. The results show that the CBHW programme in Burkina Faso offers curative care to children, targeting certain diseases. This enabled us to investigate the factors that influence confidence in the care offered and the satisfaction of the households that benefit from it. The focus group discussions enabled us to gain a better understanding of households’ perceptions of the quality of care provided by CBHWs. The use of the different dimensions described by Beneguissé enabled a true assessment of the quality of care offered by the CBHWs.

While the provision of curative care by CBHWs is a reality, the fact remains that certain aspects such as the time taken to obtain care (extended in this case by the unavailability of the CBHW at times) and interpersonal communication during the administration of care deserve particular attention, as they influence household satisfaction. The results also confirm the need to develop community-based care to meet the needs of vulnerable populations, especially those facing security and humanitarian crises. Our results allow us to suggest a more in-depth analysis of the geographical coverage and, above all, the immediate need for additional recruitment to meet the needs of households located a considerable distance from the CBHWs. Finally, solving the structural problems leading to insufficient or unavailable medicines must be a priority, as well as the introduction of innovative means and techniques of communication for CBHWs.

## Figures and Tables

**Table 1 ijerph-21-01151-t001:** Coding of quotes categories based on Beneguissé’s conceptual model of perceived quality.

Domains of Beneguissé’s Model of Perceived Quality of Care	Illustrative Quotes According to the Domain	
	Satisfaction Argument	Non-Satisfaction Argument
**Geographical accessibility**	“Yes, we are here together” Man_R3_FG12 “I’m talking about them. The villagers are chosen, they are trained and then they go back to the village to look after the children” Woman_R2_FG4 “With them we can get medicines and it saves us time because we no longer need to go to the health centre” Woman_R3_FG5 “I really agree that their work is good, because before we had to suffer a lot to get to the health centre with our sick children, and the journey itself was tiring for the patient. However, with the intervention of CBHWs, as soon as the child is ill, we go to them first and with their care they manage to relieve the patient” Woman_R6_FG5 “I really appreciate the fact that we allowed them to give us care at home. It’s a relief for us because there are times when our child is ill and we don’t even know how to get to the health centre because we don’t have the means to get there” Woman_R1_FG5	“Most of the time, when a child is ill, it is difficult to find the CBHWs at home, that is why we go directly to the health centre” Woman_R4_FG6 “Most of the time when a child is ill, it is difficult to find the CBHWs at home, that is why we go directly to the health centre" Woman_R4_FG13 “Most often, outside vaccination campaigns, CBHWs cannot be found on the spot, so it is better to go quickly with the patient to the health centre than to try to consult a CBHW” Man_R2_FG14
**Organizational accessibility**	“They are the link between the village and the CSPS (health facility), so if there is any information for the population, the health workers ask them to pass it on to us” Woman_R2_FG1 “We pay no fees for this care” Woman_R3, R4_FG6 “They don’t charge for their services” Man_R2_FG10 “We go and see the CBHWs first, and if they have medicines, they give them to us” Woman_R6_FG3 “When something goes wrong with our children, we first contact the CBHWs” Woman_R4_FG5 “It’s only when they don’t have the products, they need for treatment that we go to the health centre” Woman_R4_FG17 “They come home to see if we have really given the products to the child” Woman_R4_FG18	“When a child shows these signs of illness, my first contact is the health centre” Woman_R5_FG5 “As soon as the child doesn’t feel well, we take them to the health centre” Woman _R2_FG6 “Here, when you notice that your child’s body is hot, your first reaction is to seek advice, so straight to the hospital.” Woman_R2_FG8 “They say they don’t have all the medicines for treatment, so we have to go to the CSPS” Woman_R6_FG11
**Interpersonal communication**	“Yes, they advise us on cleanliness, and thanks to them we know that sweeping our environment, washing our kitchen utensils, and pouring waste water away from our homes are actions that can keep mosquitoes away from us and, at the same time, prevent certain diseases such as malaria and diarrhoea” Woman_R4_FG5 “They advise us to take the children for consultations as soon as we see the first signs of illness, and not to hang around the house” Woman_R1,R2,R5_FG6. “They also raise our awareness on sanitation, especially when it comes to managing domestic animals. They advise us to tie them up outside the compound because they dirty the yard and the stagnant water is a source of mosquitoes. We have to evacuate the waste water and keep the area clean” Man_R1_FG7	“We go to the CSPS because if the information about the medicines available at their level is not constantly updated, people forget, so they send their patients directly to the CSPS, if not in the past, for these illnesses they are the ones we call first” Man_R4_FG7 “CBHWs advised us not to hang around with a sick child at home, that as soon as the first signs of illness appeared, we should go immediately to the health centre” Woman_R7_FG15 “Their activities are periodic. There are times when they have medicine and when your child is ill they give it to them, but at this moment they don’t have any, so when you have an illness you have to go to the health centre quickly” Man_R3_FG10
**Technical skills**	“They gave them medicines that they keep in the village, so if your child (from 6 months to under 5 years old) is sick, you go to the CBHWs, whether male or female, who will examine the child, do (Rapid Diagnostic Test) RDTs and if the child is found to be suffering from malaria, they will give you the medicines” Woman_R1_FG1 “We’re satisfied with their work, they do a good job of care” Homme_R1_FG12 “Their care is really good, they respect us. Once my child had a fever and when he arrived, he told me to wet a towel and cover him before taking him to the health centre” Woman_R2_FG19 “Yes, our children have been treated several times; they give vitamins to the children each time, and also medicine against malaria” Man_R2_FG20 “Thanks to them, our children are treated and cured without necessarily going to the health centre” Woman_R2_FG18	“When a child is ill, I prefer to go directly to the health centre because in general the CBHWs does not have all the products needed to treat the patient properly” Woman_R5_FG5 “For me, the problem is the permanent lack of medicines in the drugs depot, so each time I have to go somewhere else to buy them” Man_R1_FG7 “There are times when they have medicine and when your child is ill they give it to them, but at this moment they don’t have any, so when you have an illness you have to go to the health centre quickly” Man_R3_FG10 “Faced with these illnesses, I quickly go to the centre because delaying can lead to complications in children” Man_R1_FG14 “In these situations, we have to bypass the CBHWs and go to the health centre because they have nothing to treat them ”Man_R4_FG14 “The CBHWs have no medicine with them, that is why we go to the health centre” Woman_R5,6,7,8_FG19
**Continuity of care**	“If the illness persists, they refer you to the centre for better care” Woman_R1_FG1 “Yes, we start there and they do RDTs and if it is malaria they give medicines and amoxicillin, and it is when the illness persists that they tell us to go to the health centre” Woman_R2_FG1 “When your child is ill and you go to see him, he treats the child and gives them medicine. But when the illness is beyond his competence, he refers you to the health centre” Woman_R3_FG3	“CBHWs only have medicines for malaria, vitamins and polio. And if we take the time to look for the CBHWs and it turns out that they don’t have any medicine, we have wasted our time for nothing. This is why we go straight to the health centre” Man_R4_FG16

**Table 2 ijerph-21-01151-t002:** Level of satisfaction on a five-point scale.

Geographical Accessibility (Distances Travelled/Existence of Natural Barriers), …)			
Rating Level	Total Number		Percentage (%)
	Very satisfied	165	17.2
	Satisfied	314	32.7
	Somewhat satisfied	219	22.8
	Not very satisfied	150	15.6
	Not at all satisfied	112	11.7
	Total	960	1.0
**Accessibility in terms of time (waiting time/scheduling of sessions, etc.))**			
	Very satisfied	171	17.8
	Satisfied	409	42.6
	Somewhat satisfied	290	30.2
	Not very satisfied	59	6.1
	Not at all satisfied	31	3.2
	Total	960	100.0
**Cultural accessibility (speaks the same language/same habits), …)**			
	Very satisfied	288	30.0
	Satisfied	326	34.0
	Somewhat satisfied	333	34.7
	Not very satisfied	13	1.4
	Not at all satisfied	0	0
	Total	960	100.0
**Courtesy when treating your child**			
	Very satisfied	248	25.8
	Satisfied	344	35.8
	Somewhat satisfied	354	36.9
	Not very satisfied	14	1.5
	Not at all satisfied	0	0
	Total	960	100.0
**Practical advice when treating your child**			
	Very satisfied	234	24.4
	Satisfied	338	35.2
	Somewhat satisfied	378	39.4
	Not very satisfied	10	1.0
	Not at all satisfied	0	0
	Total	960	100.0
**Information on illnesses in children aged 0–5 (speeches and images)**			
	Very satisfied	192	20.0
	Satisfied	381	39.7
	Somewhat satisfied	360	37.5
	Not very satisfied	26	2.7
	Not at all satisfied	1	0.1
	Total	960	100.0

**Table 3 ijerph-21-01151-t003:** Proportion of types of care provided by CBHWs reported by households.

Types of Care/Injection of Medicines			
Household Response		Frequency	Percentage (%)
	Yes	6	0.6
	No	954	99.4
	Total	960	100.0
Types of care/Give medicines to swallow			
	Yes	872	90.8
	No	88	9.2
	Total	960	100.0
Types of treatment/Give vitamins			
	Yes	702	73.1
	No	258	26.9
	Total	960	100.0
Types of care/Measures weight and arm circumference			
	Yes	426	44.4
	No	534	55.6
	Total	960	100.0
Types of care/Give advice			
	Yes	688	71.7
	No	272	28.3
	Total	960	100.0

**Table 4 ijerph-21-01151-t004:** Factors associated with household satisfaction with CBHWs care.

Factors	B	ddl	Sig.	Exp(B)	Confidence Interval 95% for Exp(B)	
					Lower	Higher
Geographical accessibility (distances travelled/existence of natural obstacles, …)		4	0.123			
Geographical accessibility (distances travelled/existence of natural obstacles, …) (1)	−1.778	1	0.114	0.169	0.019	1.535
Geographical accessibility (distances travelled/existence of natural obstacles, …) (2)	0.705	1	0.586	0.494	0.039	6.225
Geographical accessibility (distances travelled/existence of natural obstacles, …) (3)	−0.098	1	0.938	0.907	0.077	10.699
Geographical accessibility (distances travelled/existence of natural obstacles, …) (4)	−0.764	1	0.534	0.466	0.042	5.162
Accessibility in terms of time (waiting time/programming of sessions;)		4	0.001			
Accessibility in terms of time (waiting time/programming of sessions;) (1)	1.015	1	0.224	2.760	0.537	14.182
Accessibility in terms of time (waiting time/programming of sessions) (3)	15.469	1	0.998	5,223,962.022	0.000	
Accessibility in terms of time (waiting time/programming of sessions) (3)	**−2.630**	**1**	**0.005**	**0.072**	**0.011**	**0.455**
Accessibility in terms of time (waiting time/programming of sessions) (4)	−1.091	1	0.217	0.336	0.059	1.899
Cultural accessibility (speaks the same language/same habits, …)		3	0.709			
Cultural accessibility (speaks the same language/same habits, …) (1)	−0.784	1	0.313	0.457	0.100	2.092
Cultural accessibility (speaks the same language/same habits, …) (2)	−0.794	1	0.586	0.452	0.026	7.873
Cultural accessibility (speaks the same language/same habits, …) (3)	−0.362	1	0.674	0.696	0.129	3.762
Awareness-raising/practical advice on providing care for your child (treatment of malaria, diarrhoea and pneumonia)		3	0.111			
**Awareness-raising/practical advice on providing care for your child (treatment of malaria, diarrhoea and pneumonia) (1)**	**−2.554**	**1**	**0.031**	**0.078**	**0.008**	**0.790**
Awareness-raising/practical advice on providing care for your child (treatment of malaria, diarrhoea and pneumonia) (2)	−1.953	1	0.246	0.142	0.005	3.841
**Awareness-raising/practical advice on providing care for your child (treatment of malaria, diarrhoea and pneumonia) (3)**	**−2.995**	**1**	**0.015**	**0.050**	**0.004**	**0.559**
Information on illnesses in children aged 0−5 (speeches and images)		4	0.133			
Information on illnesses in children aged 0–5 (speeches and images) (1)	0.827	1	0.299	2.287	0.480	10.892
Information on illnesses in children aged 0–5 (speeches and images) (2)	19.244	1	1000	227,876,371.742	0.000	
Information on illnesses in children aged 0–5 (speeches and images) (3)	0.773	1	0.486	2.166	0.246	19.082
**Information on illnesses in children aged 0–5 (speeches and images) (4)**	**2.198**	**1**	**0.015**	**9.006**	**1.521**	**53.321**
Constant	6.500	1	0.000	665.324		

B: coefficient, ddl: degree of liberty, Sig: The degree of significance is indicated under the column, Exp: or the odds ratio, is the predicted change in odds for a unit increase in the predictor.

## Data Availability

We have collected and analysed data on our financial resources. The original contributions presented in the study are included in the article/Appendix A, further inquiries can be directed to the corresponding author/s.

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
