# Peer review of "Factors Associated with Communities’ Satisfaction with Receiving Curative Care Administered by Community Health Workers in the Health Districts of Bousse and Boussouma in Burkina Faso, 2024"

_ijerph, 2024, doi:10.3390/ijerph21091151_

Round 1

Reviewer 1 Report

Comments and Suggestions for Authors

Thank you for inviting me to review this manuscript.

The topic of patients satisfaction connected with quality is extremely important in all health care systems around the world.

The abstract is written correctly, it contains the most important issues described in the article. The methods and tools used seem to be appropriate,

The article bring much new on the issue of the need for medical care for developing countries especially African ones. The situation presented in the article is extremely interesting especially from the perspective of a resident of a European country.

The section „Concept approach, techniques and data collection tools” seems to be described in too much detail in the reviewer's opinion as well as „Data treatment, analysis and ethical aspects”. Reviewer suggests rewriting and shortening these sections

The figures properly show the data and are very detailed but rather are easy to understand. The descriptions of the results are also appropriate

References meet assumptions.

Author Response

Thanks for the comments; Our answers have been provided in the following comments

Reviewer 2 Report

Comments and Suggestions for Authors

Dear Author(s), 

Kindly see below some suggestions on how to improve the manuscript:

- please add in the study population section about the written consent of the participants as well as about the ensure of confidentiality and anonimity in data analysis

- the quality of the tool is actually the reliability of the tool and although you have mentioned the Cronbach alpha coefficient there is none in the Results section. please add one. 

- I would recommend not to include the paragraph Regression are mathematical models...neperian logarithm. This is similar to a paragraph from a text book which is not the case here. 

- why did you use the median age when we usually declare the mean age of the population. I would use the mean instead of median

- described by Beneguisse requires a reference

- table 1 I would shorten it and code it according to the repeated topics or single topics. For instance, the satisfaction of Woman R5 FG1 is the same as for Man R3 FG12 which means redundancy

- Table 2 I would rearrage the information differently- for instance rating, frequency and percentage will be repeated for each variable (geographical accessibility etc). 

- table 4- delete columns ES and Wald and use only 2 decimals after comma. 

- please add the limitations of the study in the discussion section 

- please add some suggestions for further research which would focus on the awareness of the community health workers as well as their essential role in the community. Maybe some Social Marketing campaigns will be a solution, however I am not sure taking into consideration the cultural background. 

Comments on the Quality of English Language

English language is fine. A few typos. 

Author Response

Thanks for the comments!

Our answers have been provided in the following comments.

Round 2

Reviewer 2 Report

Comments and Suggestions for Authors

English language editing is necessary!

Comments on the Quality of English Language

There are a few typos and words in English which are not approapriately used. 

For example, pag 1- line 42- born to be replaced with conceived/ created

pag 1, line 53- seen (considered)

pag 3, line 97- is is used twice